# Virological and Pharmaceutical Properties of Clinically Relevant Phages

**DOI:** 10.3390/antibiotics14050487

**Published:** 2025-05-10

**Authors:** Antonios-Periklis Panagiotopoulos, Antonia P. Sagona, Deny Tsakri, Stefanos Ferous, Cleo Anastassopoulou, Athanasios Tsakris

**Affiliations:** 1Department of Microbiology, Medical School, National and Kapodistrian University of Athens, 11527 Athens, Greece; anpanagiotop@med.uoa.gr (A.-P.P.); dtsakri@med.uoa.gr (D.T.); sferous@med.uoa.gr (S.F.); cleoa@med.uoa.gr (C.A.); 2School of Life Sciences, University of Warwick, Coventry CV4 7AL, UK; a.sagona@warwick.ac.uk

**Keywords:** phage therapy, antimicrobial resistance, lytic bacteriophages, genomic and proteomic characterization, pharmacokinetics

## Abstract

As antimicrobial resistance continues to undermine the efficacy of antibiotics, the global medical community is increasingly turning to alternative treatment modalities. Among these, phage therapy has re-emerged as a promising strategy for managing multidrug-resistant bacterial infections. Herein, we present and briefly discuss eight essential attributes of clinically relevant phages for therapy, which may be categorized broadly into virological and pharmacological characteristics. Virological attributes include a broad host range, a strictly lytic life cycle and the ability to manage the emergence of bacterial resistance to phages. Comprehensive genomic and proteomic characterization forms the foundation for selecting and engineering such candidates, ensuring both safety and predictability. From a pharmacological standpoint, phages should ideally show safety across relevant formulations and routes of administration, favorable pharmacokinetics, stability during storage and scalability in manufacturing. Advances in genomic analysis, artificial intelligence-driven phage selection and formulation technologies have further accelerated the translational potential of phage therapy. By systematically addressing each of these critical attributes, this work aims to inform the rational selection and development of therapeutic phages suitable for integration into the clinical practice.

## 1. Introduction

Bacteriophages—often referred to simply as phages—are viruses that specifically target bacterial cells and have co-evolved alongside them over billions of years [1]. Representing one of the most prevalent biological entities on the planet, phages are considered among the earliest life-associated systems to arise [2]. Their replication cycle involves infecting a bacterial host, reproducing within it and ultimately lysing the cell to release progeny that continues to infect other bacteria [3]. The presence of these bacterial viruses was first observed by Frederick Twort in 1915 and independently confirmed by Félix d’Hérelle in 1917. It was d’Hérelle who introduced the term “bacteriophage”, meaning “bacteria-eater” from the ancient Greek “*φαγεῖν*” (“phagein”), after identifying a lytic agent effective against *Shigella* [4]. Foreseeing their therapeutic potential, d’Hérelle attempted some of the earliest experimental applications of phages to treat bacterial infections in 1919 [5]. The discovery of antibiotics that proved to be effective for many years diverted scientific and clinical attention away from phage therapy in the Western world [6].

In recent years, antimicrobial resistance (AMR) has emerged as a major obstacle to the success of antibiotic therapies [6]. Alternative methods for treating infections of multidrug-resistant (MDR) bacteria have become necessary [6]. Among the most promising alternatives is phage therapy, which has been successfully utilized mostly in several Eastern European countries for nearly a century [7]. Its long-standing use is largely attributed to the natural abundance of phages and the cost-effectiveness of their production [7]. In order to apply phage therapy extensively in the clinical setting of modern “Western” medicine, we need to identify phages that have all the desirable characteristics that render them safe and effective under the stringent criteria imposed by current bioethical regulatory standards [8]. The following section examines the eight critical attributes that phages must exhibit to be considered suitable for therapeutic applications, as summarized in Figure 1. It is important to note that several of these characteristics are interrelated and certain features therefore may be relevant to multiple categories.

## 2. The Eight Essential Characteristics of Therapeutic Phages

### 2.1. Broad Host Range

One of the key advantages of phages is their remarkable specificity, which allows them to selectively target bacterial strains without disrupting the host’s natural microbiota [9]. While this specificity is highly desirable from a safety perspective, it presents a significant challenge for the clinical implementation of phage therapy [9]. Naturally occurring phages with a narrow host range require precise identification of the infecting bacterial strain before treatment, necessitating the creation of extensive phage libraries tailored to each strain [10]. Given the vast diversity and rapid evolution of bacterial pathogens, this approach is neither scalable nor practical for widespread clinical use. Moreover, the time required to identify the specific bacterial strain and the appropriate therapeutic phages, which may range from several hours to days, can delay treatment and reduce its efficacy, especially in acute infections. To overcome these limitations, therapeutic phages must ideally exhibit a broader host range, targeting multiple strains within a bacterial species [11]. This broader specificity is essential to making phage therapy a viable and efficient option in clinical settings [11].

Bacterial defense mechanisms at the adsorption stage present additional challenges for phage therapy, even when targeting susceptible bacterial strains [12]. Modifications of phage receptors or the obstruction of receptor accessibility can significantly hinder phage–host interactions, reducing therapeutic efficacy [12]. To address these limitations, two primary strategies have been proposed. The first involves the use of phage cocktails containing multiple phages that target a diverse range of bacterial strains, thereby increasing the likelihood of successful adsorption [12]. The second approach leverages phage engineering, which may be directed towards expanded recognition of bacterial receptors, thereby enhancing the phages’ ability to infect resistant bacterial populations [12]. These strategies are crucial for overcoming bacterial defenses and ensuring the reliability of phage therapy in clinical applications.

### 2.2. Strictly Lytic Life Cycle-Driven Phages

Phages are broadly classified into two groups: virulent and temperate phages. It is well established that virulent phages are the preferred choice for therapeutic applications [13]. Their primary advantage lies in their ability to rapidly lyse bacterial cells, leading to a swift reduction in bacterial populations [13]. This characteristic is particularly critical for the success of phage therapy, as it minimizes the risk of horizontal gene transfer (HGT) and the spread of resistance genes [13]. Temperate phages, which integrate their DNA into the host genome and multiply along with it before switching to the lytic cycle, can transfer unwanted genes more easily [14]. Additionally, virulent phages facilitate self-amplification, ensuring that their population increases in the presence of the target bacteria and naturally declines once the infection is eradicated [15]. This self-regulating mechanism enhances treatment efficiency while reducing the need for repeated dosing, making virulent phages the optimal candidates for therapeutic applications [15].

### 2.3. Resistance Management

Bacterial defense mechanisms against phages include receptor modification, biofilm formation, restriction–modification (R-M) systems, CRISPR-Cas systems and abortive infection (Abi) pathways [12]. To be effective in therapeutic applications, phages must possess characteristics that enable them to counter or bypass these barriers [12]. Recognition of different receptors is a critical attribute, allowing phages to target bacteria despite receptor modifications or surface alterations [16,17]. However, perhaps not all phages may be modified to target multiple bacterial hosts. Phages encoding depolymerases can degrade extracellular polysaccharides, enhancing their ability to penetrate biofilms and reach bacterial cells [18]. Regarding the counteraction against R-M systems, phages capable of mimicking bacterial methylation patterns or producing anti-restriction proteins are more likely to establish successful infections for the destruction of harmful bacteria [19]. Phages that evade CRISPR-Cas systems by carrying anti-CRISPR proteins or by mutating essential recognition sequences are also advantageous [20]. Natural phages with these traits or combinations are therefore most likely to succeed in overcoming bacterial defenses. Therefore, selecting phages with these capabilities is essential for effective therapeutic applications.

### 2.4. Genomic and Proteomic Characterization

Phages intended for therapeutic applications must undergo comprehensive genomic and proteomic characterization to ensure safety and efficacy [21]. Genomic analysis is essential for identifying harmful genetic elements, while also providing insights into phage functionality [21]. Proteomic characterization enhances our understanding by revealing the precise structural composition of phages, including their receptor-binding sites, which are critical for host specificity [21]. More importantly, proteomic studies offer valuable insights into phage–host interactions, aiding in the selection and optimization of phages for clinical use [21].

Recent studies have actively contributed to the genomic and proteomic characterization of phages with therapeutic potential. Niaz et al. characterized four novel *Schitoviridae* phages targeting uropathogenic *Escherichia coli*, providing detailed genomic and proteomic data essential for evaluating their suitability for clinical applications [22]. El-Din et al. focused on phages isolated from *Pseudomonas aeruginosa* clinical strains derived from early and chronic cystic fibrosis patients [23]. Their genomic analysis confirmed that the phages were strictly lytic and devoid of undesirable genes, such as those encoding antibiotic resistance, toxins or other virulence factors [23]. Qin et al. sequenced the genome of phi1_092033, a broad-host-range phage capable of lysing carbapenem-resistant *Acinetobacter baumannii* strains across various capsule types [24]. These recent advancements contribute to the growing body of knowledge necessary for the safe and effective application of phage therapy.

The application of artificial intelligence (AI) innovations in genomic and proteomic characterization techniques, through the development of machine learning (ML) algorithms for instance, has significantly expanded the number of characterized phages [25]. Additionally, such AI improvements in phage–host interaction predictions and simulations of phage–bacteria co-evolution have provided a more comprehensive understanding of the complexities involved in therapeutic phage applications [25]. AI tools can assist in identifying patterns and make predictions from large-scale biological data, such as nucleotide or protein sequences via “big data” analysis. These tools can detect genomic features, infer protein functions, and predict host–phage interactions with improved accuracy. For instance, the development of automated annotation tools, such as rTOOLS, has demonstrated comparable accuracy to manual methods like SEA-PHAGES while significantly improving processing speed and reducing costs [26]. Similarly, novel prediction models, such as PHPGAT, have shown superior accuracy in phage–host interaction predictions compared to existing models [27]. These technological advancements are accelerating phage research, making therapeutic applications more efficient and accessible.

### 2.5. Safe for Administration

As for all therapeutics, safety is a critical consideration in the clinical application of phage therapy. To be deemed suitable for therapeutic use, phages must not encode toxins, virulence factors or antibiotic resistance, as these elements could pose significant risks to the patient [28]. Some naturally occurring phages harbor genes that could contribute to bacterial pathogenicity by enhancing toxin production, increasing virulence or facilitating HGT, potentially exacerbating infections rather than resolving them [28]. While these concerns are traditionally associated with temperate phages that are not used in phage therapy protocols, recent studies have identified that even lytic phages might carry AMR or virulence genes [29].

Beyond the genetic safety of therapeutic phages, their interactions with the human immune system and microbiome must also be carefully considered [30]. The human body harbors a vast and diverse community of bacteria, as well as other viruses, collectively known as the microbiome and virome, respectively [31]. Both the microbiome and the virome it contains are currently thought to play a crucial role in maintaining health, with alterations or disruptions in their homeostasis mediating or contributing to pathological states [32].

As bacterial viruses, phages have the potential to infect and alter bacterial populations within the microbiome [30]. Disrupting this delicate balance could lead to unintended health consequences, including dysbiosis and the potential development of disease [30]. For example, Hsu et al. demonstrated that phage therapy not only reduced the targeted gut population but also caused cascading shifts across non-target microbial species, altering the broader microbiome ecosystem [33]. The human virome encompasses a diverse collection of eukaryotic and prokaryotic viruses, including phages, that coexist with the human host and interact with the broader microbiome across various anatomical sites.

Phage therapy, by altering bacterial community structure, has the potential to indirectly reshape the human virome, particularly the local viromes such as that of the gut, with possible downstream effects on microbial balance and host health [32]. For instance, Zuo et al. found that patients with active ulcerative colitis exhibited reduced diversity and abundance of gut mucosal bacteriophages, along with disrupted phage–bacteria correlations, suggesting a disturbed virome that may contribute to mucosal inflammation in UC pathogenesis [34]. Although specificity is an indication that beneficial microbial communities should be unaffected, there are observations of notable changes in the microbiota throughout the duration of phage therapy [35].

### 2.6. Favorable Pharmacokinetics

Phages intended for therapeutic use should possess pharmacokinetic properties that support effective distribution, persistence and activity within the human body [36]. Ideally, therapeutic phages should be able to replicate efficiently at infection sites in the presence of susceptible bacteria, allowing them to amplify their concentration locally [36]. Furthermore, phages should be resilient to physiological barriers, such as acidic pH in the stomach (for oral delivery) or immune-mediated clearance mechanisms like phagocytosis and neutralizing antibodies [30].

Nevertheless, the characterization of naturally occurring phages that infect bacteria in widely diverse environmental niches, ranging from sewage waters to the human gut, is still in its infancy. To enhance phage survival during oral delivery, encapsulation strategies such as embedding phages in low cross-linked anionic nanocellulose-based hydrogels have been developed, providing protection against gastric acidity [37]. Meanwhile, liposome encapsulation has been explored as a strategy to protect therapeutic phages from neutralizing antibodies and phagocytic clearance, enhancing their persistence in the circulation [38].

Resistance to rapid metabolic inactivation, particularly in the liver and spleen, contributes to prolonged circulation and therapeutic window [30]. Phages that demonstrate favorable biodistribution, reaching and maintaining effective concentrations at the site of infection—whether administered intravenously, orally or topically—are better suited for clinical translation [39]. For example, dry powder inhalers offer a practical and efficient method for delivering phages directly to the lungs, granting them the ability to achieve higher pulmonary concentrations compared to oral or intravenous formulations [39].

More importantly, the ability of phages to interact predictably with the host immune system so as to avoid rapid elimination is essential for sustaining therapeutic levels [40]. Studies have shown that while orally or topically administered phages often do not trigger significant antibody production, intraperitoneal administration can lead to increased IgG and IgM levels against phages [41]. The immune response to phages varies depending on the route of administration, phage type, and the duration of exposure, which may influence treatment efficacy [41]. Phages with the proposed pharmacokinetic strengths are more likely to perform consistently in vivo and across patient populations, rendering them better candidates for standardized, scalable therapeutic applications.

### 2.7. Manufacturing and Scalability

For phage therapy to be scalable and suitable for widespread clinical application, it is essential to select phages with inherent properties that support efficient manufacturing, formulation and purification [42]. Ideally, therapeutic phages should be capable of replicating to high titers in well-characterized, non-pathogenic bacterial strains that are compatible with Good Manufacturing Practice (GMP) production [43,44]. This ensures consistency and minimizes the risk of contamination from virulence or antibiotic resistance genes [43]. Additionally, phages must be robust enough to withstand downstream purification processes, such as tangential flow filtration and chromatography which are used to remove bacterial debris, endotoxins and other impurities [45,46]. Phages that are sensitive to these steps may lose viability or require more complex and costly processing [45]. Formulation stability is another critical factor: phages intended for therapeutic use should maintain infectivity in different delivery formats, such as liquids, lyophilized powders or encapsulated systems, and remain stable under varied storage conditions [44,47]. By prioritizing these characteristics during phage selection and development, manufacturers can streamline production and enhance the overall feasibility of large-scale phage therapy deployment.

### 2.8. Stability and Ease of Storage

Stability and storage are critical factors influencing the therapeutic potential of natural phages. There are several different methods of manipulating phages that have been used to improve life on the self for phage therapy products, but significant limitations still exist. Phage stability is highly dependent on the formulation type, with liquid, gel and powder forms varying significantly in their resilience [48]. For pulmonary delivery, pre-delivery stability is crucial to ensure phage viability during aerosolization [48]. Phages tend to degrade in pure water due to oxidation, which can damage their structure and genetic material [48]. To maintain stability, phages are often stored in buffered solutions like phosphate-buffered saline (PBS), which help preserve their infectious titer over extended periods, even at higher temperatures that are not typically preferred by viruses [48]. In addition to formulation type and buffering, the use of stabilizing excipients can further enhance phage stability, especially during long-term storage. For example, PEV20 combined with ciprofloxacin remained stable at 4 °C for 12 months, even without lactose as a stabilizing excipient [49]. Incorporating lactose enhanced stability at 25 °C, demonstrating potential for extended storage [49].

Phages stored in syringes at 4 °C exhibited stable titers for the first 7 days, after which a gradual decrease in titers was observed [50]. Notably, the stability of certain phages, such as PT07, was compromised more quickly in polycarbonate syringes, with titers dropping below acceptable levels after just two days [50]. However, phage stability remained unaffected when using devices like nasal douches or catheters, showing minimal changes in titers [50]. Eastern European countries have a long-standing history in the production and clinical application of phage therapy products, as detailed by Karn et al. [51]. Several commercially available formulations developed in these regions have demonstrated stability and efficacy, highlighting the success of established techniques and their potential for further optimization in the West [51].

## 3. Future Directions

Despite the long-standing use of phage therapy in Eastern European countries, its widespread adoption in the United States and the European Union remains limited, primarily due to regulatory and economic challenges. In Western regulatory frameworks, particularly those set by the Food and Drug Administration (FDA) and the European Medicines Agency (EMA), the bar for pharmaceutical approval is set high, with stringent requirements for demonstrating safety, efficacy, and favorable pharmacokinetics through a structured pipeline of evaluation stages that include in vitro characterization, animal studies and multi-phase clinical trials. One of the most substantial barriers to this process is financial: clinical trials are prohibitively expensive, limiting the advancement of many promising phage therapy initiatives [52]. Moreover, phage therapy is often regarded from a personalized medicine perspective, given the specificity of phages for individual bacterial strains [46]. This presents unique regulatory hurdles, as current frameworks are designed around fixed, standardized formulations rather than dynamic, patient-tailored treatments.

Additionally, even though the environmental safety associated with the release or elimination of phages from the human body is considered very low, there are some potential concerns around the impact of phages on microbial communities, which could lead to imbalances and the development of phage resistance [53]. These can be circumvented by the controlled release of phages and by genetic engineering, but it is wise to be taken into consideration when phage therapy is applied.

The inevitable development of bacterial resistance further complicates matters, necessitating the regular modification of therapeutic phage cocktails. This constant need for adaptation stands in contrast to the static nature of conventional pharmaceuticals and poses a significant challenge for regulatory approval processes. As we argue in this article, genetically engineered phages can help achieve more readily effective and adaptable phage-based therapeutics. However, their development introduces further regulatory complexities and cost implications, making large-scale clinical deployment rather complicated under existing approval models.

That said, countries like Georgia and Russia offer contrasting models of phage regulation that highlight possible lessons for Western frameworks. In Georgia, both ready-made and custom phage treatments are recognized as pharmaceuticals, with personalized formulations enabled through specially licensed pharmacies [51]. Russia, on the other hand, restricts personalized phage therapy and permits production only by approved state-linked manufacturers [51]. Under current EU legislation, phage therapies, particularly those based on wild-type phages, do not benefit from dedicated, streamlined regulatory pathways [8,51]. They are generally treated as standard medicinal products under Directive 2001/83/EC, meaning they must undergo the same rigorous approval processes as chemically synthesized drugs, including centralized authorization for recombinant phages classified as gene therapy medicinal products (GTMPs) [8,51]. In contrast to Georgia’s flexible approach to magistral preparations and Russia’s centralized production model, the EU’s framework remains rigid, often ill suited to personalized or rapidly adaptable phage therapies. Future policy revisions should consider integrating more flexible, tiered regulatory options that acknowledge the unique nature of phages, especially their variability and personalized application potential, without compromising safety or quality standards.

Looking ahead, the future of phage therapy will increasingly depend on advances in synthetic biology and materials science. The ability to engineer or synthesize phages with defined host ranges, enhanced stability and optimized pharmacokinetics offers a promising path to overcoming many of the limitations of naturally isolated phages [12]. Equally important is the refinement of delivery systems. While traditional formulations have relied on simple liquids or lyophilized powders, next-generation approaches, including encapsulation in nanocarriers, hydrogels or pH-responsive materials may, dramatically improve phage stability, targeted release and shelf life [54,55,56]. Investing in the design of phages, specific delivery platforms will be crucial to translating phage therapy from experimental settings to consistent, scalable clinical use.

## 4. Conclusions

Phage therapy has emerged as the main solution in the battle against AMR and the continuous ineffectiveness of antibiotics. While phages offer several distinct advantages, such as host specificity, self-amplification and minimal disruption to the microbiome, their successful clinical application requires careful selection and characterization. In this article, we highlighted eight essential attributes that define a therapeutically viable phage. Finding phages possessing all eight attributes may be difficult, yet possession of even some of these attributes in combination could be beneficial. These characteristics are anticipated to not only enhance therapeutic efficacy but also address key regulatory, manufacturing and immunological considerations of phage therapy. The integration of genomic and proteomic profiling, alongside implementing innovations in formulation and delivery, has brought phage therapy closer to mainstream clinical adoption. Moving forward, a multidisciplinary approach combining microbiology, bioinformatics and clinical science will be essential to unlock the full potential of phages as precision antimicrobials in the fight against MDR infections.

## Figures and Tables

**Figure 1 antibiotics-14-00487-f001:**
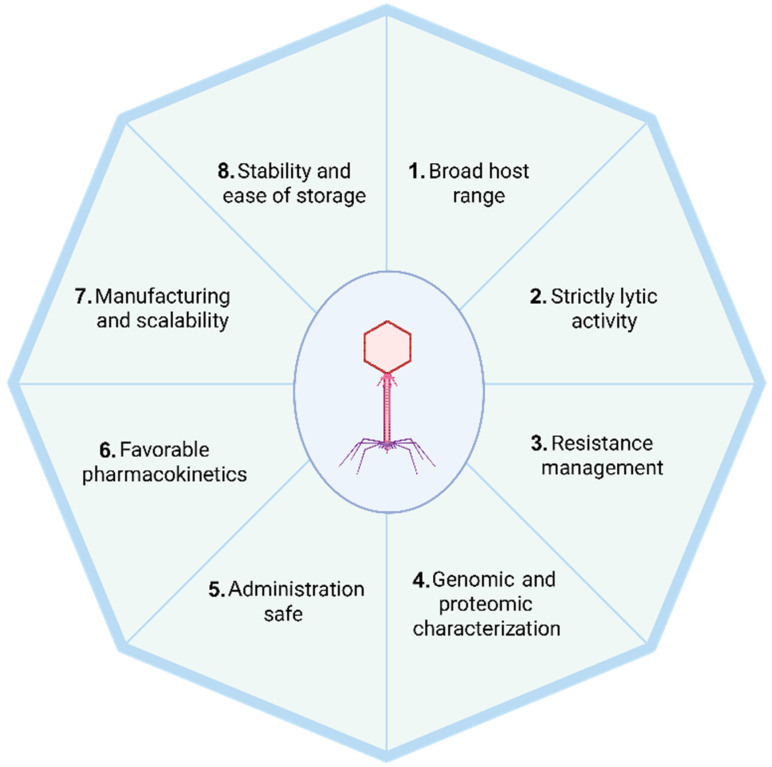
Key phage characteristics for therapeutic applications. Several of these characteristics are interrelated, and certain features therefore, such as the absence of virulence or resistance genes and the ability to self-amplify, are relevant to multiple categories. Created with BioRender.com.

## Data Availability

All of the data used in this study are included in the article.

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
