# Peer review of "Virological and Pharmaceutical Properties of Clinically Relevant Phages"

_antibiotics, 2025, doi:10.3390/antibiotics14050487_

Round 1
Reviewer 1 Report
Comments and Suggestions for Authors
The article effectively compiles key attributes of clinically significant phages into a concise shortlist. The selected attributes are well-balanced, showcasing both well-established research and emerging studies in the field. Narrowing the selection to eight attributes provides a strong foundation for discussion and encourages meaningful debate. The authors may consider the environmental safety associated with the release or elimination of phages from the human body, but this is just a minor suggestion. Recent reviews on therapeutic phages have primarily focused on clinical trials, the development of phage therapy, and associated challenges. Although there are some overlaps, this article presents a distinct and compelling perspective.
The article is concise yet comprehensive, covering numerous relevant topics, incorporating current references, and addressing key knowledge gaps. It also highlights the integration of AI and machine learning—powerful emerging tools that are driving scientific advancements forward. However, conciseness comes at the expense of illustrative examples, which could have further strengthened the discussion.
Specific comments are as follows:
Lines 16-20: The parallelism between the eight enumerated attributes and subheadings in part2 should be improved. Subheading 2.4 discusses safety from a genomic standpoint, studied through genomic and proteomic characterization. This was not included as one of the 8 attributes but mentioned in lines 20-23 as 'central to assessment and optimization'
Lines 45-46: "antimicrobial resistance (AMR) has emerged as a major obstacle…,particularly for Gram-negative bacteria". This observation is not limited to Gram negative but also for gram + infections eg. MRSA, VRE. Hence, it is inaccurate to say that AMR is a particular concern for gram negative bacteria.
Lines 50-51: Cost effectiveness was not mentioned reference [7]. Later in this article, the authors mentioned downstream processes which is relatively more expensive than the processing requirements for antibiotics.
Line 54: References for bioethical regulatory standards may have geographical variations but examples should be cited to direct readers for further reading.
Line 106-108: Additional references may strengthen the point. In the cited study [15], the phage used mas modified to target various hosts. A phage which can target 2 strains of the host was a key output from the study but is this possible for other bacteria hosts?
Line 133: Undesirable genes may be expounded. What do these genes code for and why are they to be avoided when phages are used to therapeutic use?
Line 135: What is the relevance of the finding of Qin et al.?
Line 154-159: Arguments better supported with examples, are these really concerns if there had not been evidences of such things to happen?
Line 165: Generic, cite specific studies where phages in shifted the microbiome and virome, and its effect on overall health.
Line 178-192: Are there other examples to support arguments raised eg oral delivery and resistance to stomach acid, resistance to phagocytosis and neutralizing antibodies, resistance to metabolic inactivation in the liver and spleen, etc.
Line 203-218: There are large scale production of phages, cite examples for these studies related to the arguments raised on manufacturing and scalability.
Line 230-232: The example is about stabilizers. The preceding discussions are about phase of formulation type, and buffered solutions.
Author Response
The article effectively compiles key attributes of clinically significant phages into a concise shortlist. The selected attributes are well-balanced, showcasing both well-established research and emerging studies in the field. Narrowing the selection to eight attributes provides a strong foundation for discussion and encourages meaningful debate. The authors may consider the environmental safety associated with the release or elimination of phages from the human body, but this is just a minor suggestion. Recent reviews on therapeutic phages have primarily focused on clinical trials, the development of phage therapy, and associated challenges. Although there are some overlaps, this article presents a distinct and compelling perspective.
The article is concise yet comprehensive, covering numerous relevant topics, incorporating current references, and addressing key knowledge gaps. It also highlights the integration of AI and machine learning—powerful emerging tools that are driving scientific advancements forward. However, conciseness comes at the expense of illustrative examples, which could have further strengthened the discussion.
>> We thank the reviewer for the positive feedback on our work and the constructive criticism. We agree with the reviewer on their comment about environmental safety and we have now included a sentence to showcase this in the Future Directions Section (lines 283-288). In the revised version, we provide illustrative examples which further strengthen the discussion at several instances throughout the manuscript (please see e.g. lines 177-179, 184-188 and 201-206).
Specific comments are as follows:
Lines 16-20: The parallelism between the eight enumerated attributes and subheadings in part2 should be improved. Subheading 2.4 discusses safety from a genomic standpoint, studied through genomic and proteomic characterization. This was not included as one of the 8 attributes but mentioned in lines 20-23 as 'central to assessment and optimization'
>> The text has been edited accordingly so as to improve the parallelism between the eight enumerated attributes and subheadings in part 2. In the revised manuscript Subheading 2.4 (genomic and proteomic characterization) is now included as one of the 8 attributes.
Lines 45-46: "antimicrobial resistance (AMR) has emerged as a major obstacle…,particularly for Gram-negative bacteria". This observation is not limited to Gram negative but also for gram + infections eg. MRSA, VRE. Hence, it is inaccurate to say that AMR is a particular concern for gram negative bacteria.
>> We agree and have revised the text accordingly.
Lines 50-51: Cost effectiveness was not mentioned reference [7]. Later in this article, the authors mentioned downstream processes which is relatively more expensive than the processing requirements for antibiotics.
>> Thank you for this insightful comment. This has been integrated into the revised manuscript (lines 276-282).
Line 54: References for bioethical regulatory standards may have geographical variations but examples should be cited to direct readers for further reading.
>> We agree and have added a reference to direct readers for further reading (Reference No 8). In addition, we provide examples for the varied bioethical regulatory standards in the revised version, as suggested (lines 298-314).
Line 106-108: Additional references may strengthen the point. In the cited study [15], the phage used was modified to target various hosts. A phage which can target 2 strains of the host was a key output from the study but is this possible for other bacteria hosts?
>> Thank you. We provide an additional reference to strengthen the point in the revised manuscript (Reference No. 17 in the new list) and added a phrase to indicate that such an expansion of target bacterial strains may not be feasible for all phages.
Line 133: Undesirable genes may be expounded. What do these genes code for and why are they to be avoided when phages are used to therapeutic use?
>> We have added text to indicate that undesirable genes in this context include genes encoding for antimicrobial resistance, toxins or other virulence factors. We explain why undesirable genes are to be avoided when phages are used to therapeutic use in the next section (2.5) on safety (lines 162-167).
Line 135: What is the relevance of the finding of Qin et al.?
>> We refer to the study by Qin et al. as an example of how the genomic characterization of phages can help broaden our understanding of phage biology for effectively targeting resistant bacteria. It is amongst very recent articles that contribute to the growing body of knowledge necessary for the safe and effective application of phage therapy.
Line 154-159: Arguments better supported with examples, are these really concerns if there had not been evidences of such things to happen?
>> This is a fair point raised by the reviewer since, indeed, there has been no evidence of safety concerns with the use of phage therapy. However, the application of preventive measures for phages intended for therapeutic use, which undergo rigorous genomic screening to ensure they do not encode for toxins, other virulence factors, or antimicrobial resistance prior to clinical application, ensures their safe use. We have edited the text to reflect these points in the revised manuscript (lines 164-173) and added an extra reference (No. 29 in the list).
Line 165: Generic, cite specific studies where phages in shifted the microbiome and virome, and its effect on overall health.
>> Agreed. We have incorporated specific studies demonstrating how phage therapy can shift the microbiome and virome, along with their impacts on human health, into the revised manuscript, as suggested (lines 177-179 and 184-188 and references (No. 33 and 34).
Line 178-192: Are there other examples to support arguments raised eg oral delivery and resistance to stomach acid, resistance to phagocytosis and neutralizing antibodies, resistance to metabolic inactivation in the liver and spleen, etc.
>> Indeed, there are. We added other examples in the revised manuscript (lines 199-206).
Line 203-218: There are large scale production of phages, cite examples for these studies related to the arguments raised on manufacturing and scalability.
>> Thank you for useful suggestion. We have added more references (No. 44, 46) in the revised version.
Line 230-232: The example is about stabilizers. The preceding discussions are about phase of formulation type, and buffered solutions.
>> The reviewer is correct. In the revised manuscript, we have added text to improve clarity for the reader (lines 252-253).
Reviewer 2 Report
Comments and Suggestions for Authors
The manuscript entitled "Virological and Pharmaceutical Properties of Clinically Relevant Phages for Their Therapeutic Use" offers a well-structured on the essential characteristics of therapeutic phages, especially in the context of increasing antimicrobial resistance (AMR). However, manuscript have some concepts that needs to be clarified.
Major concerns:
The article in general is more like "review" article than "persepective" article. Authors mainly discussed the characteristics of phage from several references. What lacks in the manuscript is personal or some innovative viewpoint from the authors.
While labeled a “Perspective,” the manuscript reads like a review, presenting known phage characteristics and citing numerous studies. It does not provide a personal or innovative viewpoint from the authors as expected of a perspective.
Some improvement that are needed:
1. Give perspective on, What are the major obstacles to achieving FDA or EMA approval for phage therapeutics? Based on what written in the article, the use of phage are promising for therapeutics used.
2. Could authors propose specific regulatory frameworks or compare global strategies (e.g., Georgia/Poland vs. US/EU)?
3. Opinion on future perspective on how phage can be synthesized, ethical and biosafety concerns, formulation of new delivery materials.
For Figure 1, refer directly to specific parts of the figure 1, in relevant text sections, or annotate it more explicitly to link with the “eight essential characteristics.”
Comments on the Quality of English LanguagePlease have the article proofread by native English speakers with expertise in this field
Author Response
The manuscript entitled "Virological and Pharmaceutical Properties of Clinically Relevant Phages for Their Therapeutic Use" offers a well-structured on the essential characteristics of therapeutic phages, especially in the context of increasing antimicrobial resistance (AMR). However, manuscript have some concepts that needs to be clarified.
Major concerns:
The article in general is more like "review" article than "persepective" article. Authors mainly discussed the characteristics of phage from several references. What lacks in the manuscript is personal or some innovative viewpoint from the authors.
While labeled a “Perspective,” the manuscript reads like a review, presenting known phage characteristics and citing numerous studies. It does not provide a personal or innovative viewpoint from the authors as expected of a perspective.
>> We appreciate the Reviewer’s critical feedback. We are grateful for the insightful comments and useful suggestions that have helped to improve the quality of this manuscript.
Some improvement that are needed:
1. Give perspective on, What are the major obstacles to achieving FDA or EMA approval for phage therapeutics? Based on what written in the article, the use of phage are promising for therapeutics used.
>> Thank you for the suggestion. We have expanded our discussion to include the regulatory and economic challenges associated with phage therapy approval under current FDA and EMA frameworks (Section 3. Future directions)
Could authors propose specific regulatory frameworks or compare global strategies (e.g., Georgia/Poland vs. US/EU)?
>> In the revised manuscript we compare global strategies, US/EU vs. Georgia/ Russia (lines 298-314), as suggested.
Opinion on future perspective on how phage can be synthesized, ethical and biosafety concerns, formulation of new delivery materials.
>> We now provide a concise discussion on the future role of synthetic biology in phage design, including modular engineering and we reflect on the need for investment in next-generation delivery systems in the newly added section 3 (Future directions, lines 315-324).
For Figure 1, refer directly to specific parts of the figure 1, in relevant text sections, or annotate it more explicitly to link with the “eight essential characteristics.”
>> Figure 1 has been edited accordingly and we now refer directly to the specific parts of the Figure 1, in relevant text sections of the “eight essential characteristics.”
Comments on the Quality of English Language: Please have the article proofread by native English speakers with expertise in this field
>> The article has been proofread by native English speakers with expertise in this field. Thank you.
Reviewer 3 Report
Comments and Suggestions for Authors
In their “Perspective” Panagiotopoulos et al. describe eight attributes that they consider essential for bacteriophages in order for them to be suitable for clinical applications. The manuscript is written well, it was a pleasure to read it. In scientific terms, this author did not find any fault with the manuscript and thus the recommendation is to publish it as is.
To find bacteriophages that might have all eight attributes described by the authors might be difficult, but not all of them might be equally relevant. A bacteriophage that is lytic and does not encode any toxins would be useful even if it would not survive freeze drying. Also, the authors consider GMP production as mandatory, but do not consider the production costs that would go along with such a mandate. The current “Belgian experiment” shows us that GMP might not be necessary for efficient and safe clinical applications and all historical evidence also suggests the same. However, these are just ideas of the reviewer and do not need to be reflected in this manuscript.
Author Response
In their “Perspective” Panagiotopoulos et al. describe eight attributes that they consider essential for bacteriophages in order for them to be suitable for clinical applications. The manuscript is written well, it was a pleasure to read it. In scientific terms, this author did not find any fault with the manuscript and thus the recommendation is to publish it as is.
To find bacteriophages that might have all eight attributes described by the authors might be difficult, but not all of them might be equally relevant. A bacteriophage that is lytic and does not encode any toxins would be useful even if it would not survive freeze drying. Also, the authors consider GMP production as mandatory, but do not consider the production costs that would go along with such a mandate. The current “Belgian experiment” shows us that GMP might not be necessary for efficient and safe clinical applications and all historical evidence also suggests the same. However, these are just ideas of the reviewer and do not need to be reflected in this manuscript.
>> Thank you for your positive comments on our “Perspective.” We agree with the ideas of the reviewer and have thus added: (1) more text to indicate that it would be difficult to find phages with all eight attributes that may not all be equally relevant (lines 331-332); and (2) a brief discussion about the production costs associated with GMP (lines 294-297).
Round 2
Reviewer 2 Report
Comments and Suggestions for Authors
Thank you for addressing the comments, I think the article is better than previous one. I wish good luck for the next step.